# Mitochondrial Dysfunction: Effects and Therapeutic Implications in Cerebral Gliomas

**DOI:** 10.3390/medicina60111888

**Published:** 2024-11-18

**Authors:** Gerardo Caruso, Roberta Laera, Rosamaria Ferrarotto, Cristofer Gonzalo Garcia Moreira, Rajiv Kumar, Tamara Ius, Giuseppe Lombardi, Maria Caffo

**Affiliations:** 1Unit of Neurosurgery, Department of Biomedical and Dental Sciences and Morphofunctional Imaging, University of Messina, 98125 Messina, Italy; roberta.laera@studenti.unime.it (R.L.); rosamaria.ferrarotto@studenti.unime.it (R.F.); cristofer.garciamoreira@studenti.unime.it (C.G.G.M.); maria.caffo@unime.it (M.C.); 2Faculty of Science, University of Delhi, New Delhi 110007, India; rajivkumar@nimsuniversity.org; 3Neurosurgery Unit, Head-Neck and NeuroScience Department, University Hospital of Udine, 33100 Udine, Italy; tamara.ius@gmail.com; 4Department of Oncology, Oncology 1, Veneto Institute of Oncology IOV, Istituto di Ricovero e Cura a Carattere Scientifico (IRCCS), 35128 Padua, Italy; giuseppe.lombardi@iov.veneto.it

**Keywords:** glioblastoma, glioma, gliomagenesis, mitochondrion, oxidative phosphorylation

## Abstract

Gliomas are the most common primary brain tumors, representing approximately 28% of all central nervous system tumors. These tumors are characterized by rapid progression and show a median survival of approximately 18 months. The therapeutic options consist of surgical resection followed by radiotherapy and chemotherapy. Despite the multidisciplinary approach and the biomolecular role of targeted therapies, the median progression-free survival is approximately 6–8 months. The incomplete tumor compliance with treatment is due to several factors such as the presence of the blood–brain barrier, the numerous pathways involved in tumor transformation, and the presence of intra-tumoral mutations. Among these, the interaction between the mutations of genes involved in tumor bio-energetic metabolism and the functional response of the tumor has become the protagonist of numerous studies. In this scenario, the main role is played by mitochondria, cellular organelles delimited by a double membrane and containing their own DNA (mtDNA), which participates in numerous cellular processes such as the regulation of cellular metabolism, cellular proliferation, and apoptosis and is also the main source of cellular energy production. Therefore, it is understood that the mitochondrion, specifically its functional alteration, is a leading figure in tumor transformation, including brain tumors. The acquisition of mutations in the mitochondrial DNA of tumor cells and the subsequent identification of the so-called mitochondria-related genes (MRGs), both functional (mutation of Complex I) and structural (mutations of Complex III/IV), have been seen to play an important role in metabolic reprogramming with increased proliferation, resistance to apoptosis, and the progression of tumorigenesis. This demonstrates that these mitochondrial alterations could have a role not only in the intrinsic tumor biology but also in the extrinsic one associated with the therapeutic response. We aim to summarize the main mitochondrial dysfunction interactions present in gliomas and how they might impact prognosis.

## 1. Introduction

Gliomas are a heterogeneous group of tumors, arising from the glial cells located in the central nervous system (CNS). They are the most common primary brain tumor. Based on a statistical report of CNS primary tumors diagnosed in the USA from 2014 to 2018, every year, there are about 80,000 to 90,000 newly diagnosed cases of primary brain tumors, approximately 25% of them being gliomas [1]. The total number of glioblastoma (GB) cases diagnosed each year is about 12,000 (approximately 15% of the total of newly diagnosed brain tumors and roughly 50% of all malignant brain tumors) [1].

Gliomas have a wide range of prognoses: from low-grade pilocytic astrocytomas to highly malignant GBs. Currently, they are graded according to the 2021 World Health Organization (WHO) grading system, based on molecular and genetic markers [2]. The new WHO classification also evaluates biomolecular data which, together with histological data, allow us to obtain a more specific classification of glial tumors [2]. The advances in molecular and genetic research have led to an improved diagnostic accuracy and the development of new therapies. But, despite these advancements, the high-grade gliomas and their highly infiltrative behavior still determine a very bad prognosis. The neurological examination of these patients can range from normal to different degrees of focal weakness, sensory deficits, or, in a more severe situation, an altered mental status due to a mass effect resulting from peritumoral edema. The symptoms may vary about the location of the lesion. But, although the diagnosis is made in symptomatic patients, there is a small percentage (about 3–10%) of cases diagnosed incidentally (e.g., after a traumatic brain injury or during work-up for other conditions) [3].

The standard treatment involves surgical resection, followed by radiation therapy combined with chemotherapy with temozolomide (TMZ) and then TMZ alone. The recent advances in glioma molecular pathology and biology have provided evidence of the various genes involved in cell growth, apoptosis, and angiogenesis. The modulation of gene expression at more levels, such as DNA, mRNA, proteins, and transduction signal pathways, may be the most effective modality to down-regulate or silence some specific gene functions [4].

One of the main hallmarks of carcinogenesis is the ability to evade apoptosis and identify metabolic pathways that allow the production of energy (adenosine triphosphate or ATP), making the tumor cell capable of surviving, reproducing, and developing a resistance profile [5,6]. These functions allow mitochondria to perceive cellular stress conditions and to respond to them by ensuring adequate adaptation plasticity. It has been observed that, in a tumor cell, there are not only mutated mitochondria (mtDNA mutations) but also non-mutated mitochondria (wild-type mtDNA) determining a condition of heteroplasmy [6,7]. This condition is fundamental in the process of carcinogenesis because it allows, on the one hand, guaranteed rapid tumor adaptation, increasing the degree of cellular fitness, and, on the other hand, it would seem to participate in chemotherapy resistance by developing in that mutated mitochondrial subpopulation an altered functional polyhedral framework. Various alterations of the mitochondrial activities have been found in gliomas, such as structural and functional changes, affecting mitogenic, hemodynamic, bioenergetic, and apoptotic signaling.

This study aims to review the role of mitochondria in the development and progression of gliomas, also hypothesizing their role as a possible future therapeutic target.

## 2. Gliomas

The diffuse gliomas in adults can be divided into three groups: glioblastoma isocitrate dehydrogenase (IDH)-wild-type, astrocytoma IDH-mutant, and oligodendroglioma IDH-mutant, with 1p/19q deleted [2]. The significant mutations commonly found in gliomas include: IDH1/IDH2 mutations present in lower-grade gliomas and secondary GBs and associated with better prognosis; telomerase reverse transcriptase (TERT) promoter mutations common in primary GBs and associated with poor outcomes; 1p/19q co-deletion demonstrated in oligodendrogliomas and associated with better treatment responses; O-6-methylguanine-DNA methyltransferase (MGMT) promoter methylation, a biomarker for a better response to temozolomide (TMZ) chemotherapy. These genetic factors are crucial for glioma diagnosis and personalized therapeutic strategies [8].

### Therapeutic Strategies

To date, the treatment of malignant gliomas represents a difficult challenge due to their cellular and molecular heterogeneity. In the management of these patients, several important factors must be considered, such as the histotype of the lesion, location and volume, the extent of surgical resection and residual volume, Karnofsky Performance Score, and the positivity of specific markers [9]. Furthermore, the efficacy of the current chemotherapeutic treatments is limited by the presence of the blood–brain barrier (BBB) and the presence of genetic mutations (IDH1, PTEN, and 1p19q codeletion), which can interfere with the response to these treatments. Gliomas, particularly high-grade gliomas, consist of multiple subtypes of tumor cells with diverse genetic profiles, making it difficult to target the tumor comprehensively with a single therapy [10]. Epigenetic modifications, enzymes, and noncoding RNAs are often cell-type specific and can aid in the identification of the cell of origin. Koso et al. found evidence that the cell of origin in some GBs is an astroglia-like cell, and that the originating mutations can occur in neural stem cells [11]. The immunosuppressive tumor microenvironment of gliomas, characterized by the presence of regulatory T cells, tumor-associated macrophages, and microglia, reduces the efficacy of immunotherapy and complicates the treatment efforts [12].

The most commonly adopted treatment of gliomas is multimodal, including surgical resection, radiotherapy, and chemotherapy. The improvements in surgical techniques, including the intra-operative mapping of the eloquent areas and the use of fluorescent dyes that are helpful in the detection of tumor borders, may only offer some benefits in prolonging survival. Surgical treatment is invasive but represents the first choice for gliomas due to the difficulties arising in early-stage detection. However, radical resection is not always achievable, both due to the extensive infiltration of the tumor and as an attempt to preserve the functional areas. Furthermore, after surgical treatment, neoplastic cells tend to recur immediately, and in 95% of cases, the recurrence begins from the areas adjacent to the margins of the surgical resection [13,14]. Furthermore, when the tumor is in deep structures (diencephalon, midbrain, corpus callosum) it is preferable to perform a biopsy to obtain a histological diagnosis of the tumor. Radiation therapy and chemotherapy are non-invasive options often used as an adjuvant therapy but may also be effective for curing early-stage tumors. Radiotherapy is burdened by important side effects such as post-radiation leukoencephalopathy, nerve damage, hair loss, vomiting, infertility, and skin rashes [15]. The BBB remains a significant obstacle for many therapies, including chemotherapeutics and targeted therapies, as it restricts the penetration of large molecules into the brain [13,15]. BBB is a physical barrier that protects the brain from the passage of drugs, neurotoxins, and invading organisms and regulates the passage of nutrients between the systemic circulation and the brain. Most of the chemotherapeutic drugs used have reduced solubility, are combined with toxic solvents, and, showing a reduced specificity for the targets, can also cause damage to healthy cells.

Gliomagenesis is a multifactorial process involving a large number of molecules. ECM proteins, proteases, cell adhesion molecules, and their related signaling pathways show a vital role in glioma proliferation [16]. The blockage of activation of oncogenic pathways, either at the ligand receptor interaction level or by inhibiting the downstream signal transduction pathways, could slow down tumor progression. Approximately 50% of GBs show amplification or mutation in the epidermal growth factor receptor (EGFR), making this a promising target for therapies such as tyrosine kinase inhibitors (TKIs) [17]. Some gliomas, especially pediatric gliomas, may harbor proto-oncogene B-Raf (BRAF) mutations. Inhibitors targeting the BRAF V600E mutation are being explored [18]. The phosphoinositide 3-kinase (PI3K)/protein kinase B (PKB or AKT)/mammalian target of the rapamycin (mTOR) pathway is often dysregulated in gliomas, leading to unregulated cell growth and survival. Inhibitors targeting this pathway, such as (mTOR) inhibitors, are under investigation [19]. The complexity and crosstalk between signal transduction pathways limit the potential efficacy of targeting a single receptor or molecule. Since single-agent therapies have shown no significant benefit, it is critical to begin designing rational combinations. There is a need for combinatorial approaches or novel delivery mechanisms to overcome the resistance to and enhance the effectiveness of these therapies.

Immunotherapy is becoming a prominent focus in glioma research, particularly for high-grade gliomas. Oncolytic virotherapy using engineered viruses, chimeric T cell receptor (CAR)-T therapy targeting glioma-specific antigens like epidermal growth factor receptor (EGFRvIII), and bispecific antibodies represent the innovative immunotherapeutic strategies. Despite some promising early-phase results, these approaches still require further investigation to improve their efficacy [20]. Each of these approaches represents efforts toward more personalized and effective glioma treatment strategies. Techniques like suicide gene therapy (which delivers genes to tumor cells that trigger their death) and tumor suppressor gene therapy are being explored to provide more targeted treatment. Even their clinical efficacy is still under evaluation [21]. The advancement of gene-editing tools like clustered regularly interspaced short palindromic repeats/protein 9 (CRISPR/Cas 9) [22] offers new possibilities for correcting the genetic mutations associated with glioma growth. Epigenetic therapies, such as drugs targeting DNA methylation or histone modifications, are also being investigated to modulate the gene expression in glioma cells. However, these strategies are still in their early research phases [23]. Numerous studies have also highlighted the pivotal role of mitochondria-related genes (MRGs) in the initiation and progression of GB. However, the specific contributions of MRG coding proteins to GB pathology remain incompletely elucidated. The identification of prognostic MRGs in GB holds promise for the development of personalized targeted therapies and the enhancement of patient prognoses.

Vaccine therapies for gliomas aim to stimulate the immune system to recognize and attack tumor cells. Vaccines based on specific glioma-associated antigens, like the EGFRvIII mutation, are being tested in clinical trials. Although the early results are promising, there are a lot of trials still ongoing to prove if there is a significant improvement in overall survival [24].

An emerging approach in glioma treatment is theranostics, which combines diagnostic and therapeutic capabilities into a single platform. For instance, nanoparticles can be engineered to deliver therapeutic agents while simultaneously allowing for the imaging of tumor progression or treatment response. This personalized medicine approach holds the potential to improve glioma management [25].

## 3. Mitochondrial Activity

Mitochondria are cytoplasmic organelles that perform various crucial functions in eukaryotic cells. They are considered energy centers in cells [26]. Mitochondria are delimited by a double membrane: the external one allows the passage of small molecules, and the internal one is selectively permeable and is made up of numerous coils, recesses, and protrusions called mitochondrial cristae [26]. The function of these structures is to increase the membrane surface in which are placed a greater number of ATP complexes to provide greater energy [27] (Figure 1). The mitochondrial matrix contains numerous enzymes, ribosomes (70S, smaller than those present in the rest of the cell), and double-stranded circular DNA molecules. It is essential for cellular respiration and mitochondrial fatty acid synthesis (mtFASII).

### 3.1. ATP Production

The main function of mitochondria is the production of energy to ensure the biomolecular functions in the cell. The mitochondrial respiratory chain is the basic structure for oxidative phosphorylation [26,27]. It is composed of four enzymatic complexes, located in the inner mitochondrial membrane: nicotinamide adenine dinucleotide ubiquinone reductase (NADH dehydrogenase, Complex I, CI), succinate ubiquinone oxidoreductase (Complex II, CII), ubiquinone cytochrome oxidoreductase (Complex III, CIII), and cytochrome c oxidase (Complex IV, CIV), and two mobile electron carriers, ubiquinone (Co Q) and cytochrome c (Cyt c) [28] (Table 1). The main source of energy in cells comes from the dephosphorylation of an ATP molecule into an adenosine diphosphate (ADP) molecule. ATP production involves two molecular pathways: glycolysis, which occurs in the cytosol and is oxygen-independent, which produces two molecules of ATP and two molecules of pyruvate and the citric acid cycle (or tricarboxylic acid cycle or Krebs cycle) which occurs in the mitochondria and is oxygen-dependent [6,29,30]. This cycle is composed of nine different enzymatic reactions, and each molecule of the pyruvate cycle generates three molecules of nicotinamide adenine dinucleotide (NADH), a flavin adenine dinucleotide (FADH2), and guanosine triphosphate (GTP), a process known as oxidative phosphorylation. This process involves the transfer of electrons from NADH and FADH2 to oxygen via the electron transport chain (ETC). This oxygen-dependent mechanism is called cellular respiration. Respiration is coupled to the production of ATP-by-ATP synthase or complex V of the ETC. NADH binds to complex I (CI, NADH dehydrogenase) and is oxidized to NAD+, donating two electrons to a flavin mononucleotide (FMN) inserted in the CI subunit [28,31]. The electrons are transferred to the oxidized form of coenzyme Q or ubiquinone (Q), which absorbs two protons to form ubiquinol (QH2). When electrons are transferred from one redox center to the other, four protons are pumped through CI out of the matrix. FADH2 is derived from the oxidation of succinate to fumarate by complex II (CII, succinate dehydrogenase) during the Krebs cycle, and two electrons are transferred to Q. This process increases the ubiquinol pool but does not directly affect the proton gradient because CII is not a proton pump. CIII (Q-cytochrome c oxidoreductase) oxidizes QH2 to Q and passes the electrons to another soluble carrier, cytochrome c, in a process known as the Q cycle [28,31]. The final steps of oxidative phosphorylation occur in CIV (terminal oxidase), which allows electrons to pass from cytochrome c to oxygen, producing water. Since cytochrome c carries only one electron, four molecules are oxidized to generate two H_2_O molecules from one O_2_ molecule. Meanwhile, four protons from the substrate are taken from the matrix to form H_2_O, and the other four protons are pumped into the intermembrane space [6,29,30].

### 3.2. Apoptosis

Programmed cell death or apoptosis occurs via two signaling pathways: the extrinsic pathway in which pro-apoptotic messages come from outside of the cell and the intrinsic pathway, which requires mitochondrial outer membrane permeabilization [32,33]. Both the extrinsic and intrinsic pathways have in common the activation of the central effectors of apoptosis, a group of proteases called caspases, which direct the destruction of the structural (cytoskeleton) and functional (organelles) elements of the cell.

#### 3.2.1. Extrinsic Pathway

Tumor necrosis factor (TNF) is a protein comprising 157 amino acids produced mainly by macrophages and is the main activator of the extrinsic apoptosis pathway. This factor binds two receptors present on the cellular membrane, Rtnf1 and Rtnf2, inducing the activation of caspases. The FAS (Apo 1 or CD95) receptor is another receptor for extrinsic apoptotic signals and belongs to the TNF receptor superfamily [33,34]. FAS is a transmembrane protein that has a DD (death domain) devoid of catalytic activity. Their interaction causes the binding of adapter proteins such as FADD (FAS-associated protein with death domain) and TRADD (tumor necrosis factor receptor type 1-associated death domain protein). These proteins have an N-terminal extremity to which procaspases 8 and 10 are linked. The ligand–receptor interaction induces the proteolytic cleavage of procaspases and the creation of caspases 8 and 10 [34].

#### 3.2.2. Intrinsic Pathway

The intrinsic pathway is activated by signals internal to the cell or by signals of an extracellular origin. A fundamental role in this pathway is played by the proteins of the Bcl2 (C-cell lymphoma 2) group which regulate the mitochondrial permeability. The Bcl2 family is composed of transmembrane proteins localized on the outer mitochondrial membrane that can have both pro-apoptotic (Bax, Bak, Bok, Bid, Bim, Bad, Noxa, and Puma) and antiapoptotic functions (Bcl-xL, Bcl-w, A1, and Mcl-1) [33,35]. In mammalian cells, Bax is predominantly localized in the cytosol. Only a small amount is found on the mitochondrial surface. Bcl-xl takes care of bringing Bax back into the cytosol to avoid its accumulation at the mitochondrial level and consequent self-activation. The BH3-only class of proteins contains a BH3 domain and amphipathic helix responsible for the interaction with the Bcl-2 family members. Most of the BH3-only proteins translocate to the mitochondrial outer membrane, upon death stimuli, to bind to pro-apoptotic Bcl-2 family members (Bax and Bak, after their oligomerization in the mitochondrial surface) [33,34]. When cell survival signals prevail, proapoptotic transmembrane proteins (Bax, Bak) are bound and inhibited by antiapoptotic factors including Bcl-2. However, if the pro-apoptotic factors Bax and Bak prevail, they form oligomers capable of directly or indirectly inducing the release into the cytosol of factors capable of triggering apoptosis including cytochrome c, SMAC/DIABLO, AIF (a factor which induces apoptosis), and endonuclease G. Cytochrome c initiates apoptosis when released from mitochondria into the cytosol. Once released, cytochrome c binds to Apaf-1 (apoptotic peptidase activating factor 1). Further stabilization and binding of ATP to the Apaf-1/cytochrome c complex results in the oligomerization and formation of the apoptosome. This multimeric complex exposes the CARD domains of Apaf-1, resulting in an open conformation able to bind and activate procaspase-9 forming the active apoptosome [35]. Caspase 9 can now cleave and activate the downstream executioner caspase-3 which starts a molecular cascade which ends with the degradation of DNA by nuclear factors [33,36].

### 3.3. Mitochondrial Dynamics and Retrograde Signaling

Mitochondria are connected by a dynamic network that can undergo structural alterations in a process termed “mitochondrial dynamics” that includes fusion and fission processes [37]. Mitochondrial fusion is mediated by three GTPases of the dynamin superfamily: Mitofusin 1 (Mfn1), Mfn2, and Optic Atrophy 1 (Opa1). Those proteins allow a two-step process requiring outer-membrane fusion followed by inner-membrane fusion and have the role of increasing the connection between mitochondria, favoring the exchange of proteins, metabolites, and mitochondrial DNA. Mitochondrial fusion is vital as it allows the enhancement of their overall respiratory function [38]. Fission leads to the fragmentation of mitochondria, mediated by the dynamin-related protein 1 (Drp1), a large GTPase that moves to the outer mitochondrial membrane of damaged and dysfunctional mitochondria, where it assembles to form a multimeric ring that induces mitochondrial division and fragmentation allowing the removal of non-functioning mitochondria through a mechanism called “mitophagy” [39]. The balance between the processes of mitochondrial fission and fusion is fundamental for the correct functioning of mitochondria. The excess of or reduction in both can affect the functionality of the mitochondria, causing various pathologies including cerebrovascular diseases and tumors. Mitochondrial dynamics is central in gliomagenesis. Mitochondrial division (fission) and one of its central effectors, dynamin-related protein 1 (Drp1), have been observed to be enhanced in gliomas and involved in migration and invasiveness [40].

Mitochondrial retrograde signaling is a peculiar flow of data between the mitochondrion and the nucleus that occurs even in conditions of altered mitochondrial functionality. Generally, the loss of membrane potential, due to the alterations of the respiratory chain components or to the mutations and/or alterations of mtDNA, triggers mitochondrial retrograde signaling. This mechanism should allow the restoration of the correct mitochondrial functionality [41].

## 4. Mitochondrial Dysfunctions in Gliomagenesis

One of the main hallmarks of carcinogenesis is the ability to evade apoptosis and identify the metabolic pathways that allow the production of energy making the tumor cell capable of surviving, reproducing, and developing a resistance profile. These mechanisms are controlled at different levels in the cell [5,6]. One of the organelles involved in this sense is the mitochondrion, an organelle that contains its own DNA (ds-c-mtDNA), which encodes 37 genes: 13 polypeptides of the mitochondrial respiratory chain (complex I, III, IV, V), 22 tRNAs, and 3 rRNAs necessary for mitochondrial protein synthesis. Mitochondria functions ensure that mitochondria can perceive the conditions of cellular stress and can respond to them by providing adequate adaptation plasticity. This does not only occur in para-physiological conditions, such as in a classic inflammatory response, but above all in those conditions in which the cell undergoes a tumor transformation.

The process of gliomagenesis involves the alterations of several pathways including mitochondrial ones. In the early stages of gliomagenesis, at the mitochondrial level, molecular alterations occur, such as point mutations, deletions, insertions, and microsatellite instability [42]. In this sense, at the level of the D-loop, a non-coding mitochondrial nucleotide region involved in the regulation of transcription and molecular replication, in the polycytosine (poly-C) mononucleotide repeat tract located between 303 and 315 nucleotides known as D310, has been identified as a hot spot region for somatic mutations of mtDNA in various tumor lines, including brain ones. This region is involved in the repair of double strand breaks in DNA. Mutational phenomena affecting that region would play a role in tumor progression [42]. Szmyd et al. in a study, re-evaluated the role of the D-loop mutation, specifically of the m.16126T>C variant, which is associated, in patients with GB, with low overall survival, probably having a prognostic role [43].

In tumor transformation, there is an alteration of the conditions of the intracellular and inter-cellular microenvironment. In this sense, Warburg described the principle according to which tumor cells are able to modify their energy metabolism: tumor cells, instead of using an oxidative metabolism, prefer the glycolytic pathway, even in conditions of normal oxygen percentage [44]. In this phenomenon, which is called metabolic reprogramming, mitochondrial dysfunction is observed: 1—an excellent yield is obtained at a low cost (one molecule of glucose: two molecules of ATP, NADH, and pyruvate), 2—the catabolic and anabolic pathways of glucose are diverted into pathways that guarantee amino acids and lipids for tumor differentiation and growth; 3—an acidic microenvironment is produced that facilitates tumor infiltration and dissemination. This mitochondrial dysfunction is also observed in conditions of the normal activity of the organelles, showing in this sense, that this plasticity, “temporarily”, manifests itself from the beginning of carcinogenesis. In this sense, it seems that the translocator protein located on the external mitochondrial membrane (TPSO) interferes with the type of tumor cell metabolism. Fu et al. demonstrate that TPSO is involved in mitochondrial dysfunction and mitochondrial enhancement, as a reduction in this protein limits oxidative phosphorylation favoring glycolysis, improving the glucose uptake, and increasing the pro-angiogenic aspect. Therefore, low levels of TPSO promote the Warburg effect, establishing a pro-tumor microenvironment [45].

A cell, whether healthy or tumorous, can contain numerous mitochondria and it has been observed that, in a tumor cell, there are not only mutated mitochondria (mtDNA mutations) but also non-mutated mitochondria (wild-type mtDNA) determining a condition of heteroplasmy [46]. This condition is fundamental in the process of carcinogenesis because it allows, on the one hand, guaranteed rapid tumor adaptation, increasing the degree of cellular fitness, contrary to a condition of homoplasmy, while on the other hand, it would seem to participate in chemotherapy resistance by developing in that mutated mitochondrial subpopulation an altered functional multifaceted framework [46]. The ability of tumor cells to respond both to changes in the micro-environment and chemotherapy treatments lies in the fact that the pool of tumor cells is not univocal, the so-called inter-tumor heterogeneity, identifying both the so-called tumor-initiating cells (TICs) and cells with different degrees of differentiation (Table 2). It is precisely the TICs, “the tumor seed”, that is responsible for the survival of the tumor and therefore the development of resistance pathways. This tumor subpopulation presents bioenergetic or glycolytic-dependent or O_2_-dependent pathways, or even both profiles may be present in a single tumor population [6,7]. In GB, a condition of hypoxia reigns, given by the imbalance between the number of tumor cells and the degree of neoangiogenesis. This oxygen-poor microenvironment promotes a more aggressive tumor phenotype. It has been seen that this condition determines [47] the overexpression of CD133 of the TICs responsible for the resistance to cisplatin, temozolomide, and etoposide. Furthermore, the hypoxic condition confers a degree of radioresistance, as low O_2_ levels produce low levels of free radicals, reducing the DNA damage [48].

The mitochondrion participates in and regulates programmed cell death or apoptosis. In this sense, the process of permeabilization of the outer mitochondrial membrane favors the movement of proteins, such as cytochrome C and Smac, into the cytoplasm with the activation of caspases and therefore, caspase-mediated apoptosis. This phenomenon is regulated by the BCL-2 family, including antiapoptotic proteins (e.g., BCL-2, BCL-xL, and MCL-1) and pro-apoptotic proteins such as BAX and BAK proteins and BCL-2 homology domain-3 (BH3)-only proteins such as BIM and NOXA that contain a single BH3 domain [33,49]. It has been reported that antiapoptotic BCL-2 proteins are expressed at high levels in malignant gliomas. High levels of BCL-xL expression are associated with poor progression and survival of GB patients, and BCL-xL has been proposed as a marker of GB chemoresistance. At the therapeutic level, BH3 mimetics represent a class of inhibitors that mimic the activity of BH3-only proapoptotic proteins. For example, ABT-199 selectively inhibits BCL-2, while ABT-263 antagonizes both BCL-2 and BCL-xL [50]. The BCL-2/BCL-xL inhibitor, ABT-737, has been reported to induce apoptosis in glioblastoma cells both in vitro and in vivo and to increase the sensitivity to chemotherapeutic agents and TRAIL receptor ligation [51]. Furthermore, ABT-737 in combination with ionizing radiation induces apoptosis in glioblastoma cells, which is inhibited by p53 Wt and counteracted by p53 Mt through high levels of MCL-1 [49].

At the bioenergetic level, a mitochondrial modification as a tumor response lies in the functionality of the C1 complex of the electron transport chain. The C1 complex has the function of converting NADH into NAD+, generating both a transmembrane potential necessary to produce ATP and reactive oxygen species (ROS) [52]. In fact, according to Santidrian et al., an up-regulation or overfunction of the complex is associated with a reduction in tumor growth, and low levels of NAD+ can promote an aggressive and metastatic phenotype in breast cancer cells; therefore, mutations affecting the proteins that constitute the C1 complex participate in tumor progression [53]. A study conducted by Gasparre et al. proposed the hypothesis of a “heteroplastic threshold”, according to which the mitochondrial pool of tumor cells must have a threshold level of mutations below which it would guarantee a pro-tumor profile, while above this threshold a mutational excess would be reached that would not promote tumor progression (the oncogenes effect) [54,55].

## 5. Mitochondria as a Possible Target

From what has been reported above, we can affirm that mitochondria have an active role in the processes of tumor initiation and progression. The production of ROS by mitochondria is significantly increased in tumor cells, as they require more energy for intense metabolic activity. In addition, the conversion of aerobic glycolysis, by inducing hypoxia, promotes the proliferation of neoplastic cells. Therefore, the idea of considering mitochondrial activity as a possible target in new therapeutic protocols in the treatment of brain gliomas certainly appears plausible (Table 3).

Experimental studies have demonstrated the primary role of tumor hypoxia and tumor-initiating cells (TICs) in tumor recurrence and the resistance to radio and chemotherapy [56]. TICs seem to be closely connected to mitochondrial activity, and therefore, it seems plausible to hypothesize the use of drugs capable of limiting mitochondrial activity and, at the same time, correcting the ischemia present in the tumor microenvironment. Atovaquone is an anti-malarial drug capable of inhibiting complex III (cytochrome bc1 complex). In tumor cell lines, the use of this drug has shown a reduction in the ischemic phenomena and a reduction in the oxygen consumption rate [57]. A recent experimental study hypothesized the importance of LON, an ATP-stimulated protease, in the progression processes of glial tumors. The authors demonstrated the key role of LON in glioma cell hypoxic survival and mitochondrial respiration. The study demonstrated that in glioma cell lines and malignant glioma tissue, an increase in LON levels correlated with greater malignancy. The reduction in LON levels caused a drastic decrease in tumor cell proliferation under hypoxic conditions [58].

Recently, Sumiyoshi et al. documented the synergistic action of the pharmacological compound menadione/ascorbate on tumor cell lines and in GB mice models. The investigators demonstrated reduced tumor growth and increased oxidative stress with concomitant decreased reducing capacity. The pharmacological compound would have stimulated specific cytotoxicity and increased the mitochondrial superoxide generation only in GB cells [59]. The Bcl-xL protein plays a key role in the control of cell death by inhibiting apoptosis. Researchers have shown, on glioma cell lines, that the survivin inhibitor, YM155 was able to inhibit the activity of Bcl-xL. This protein can likely induce the loss of mitochondrial membrane potential, release of cytochrome c, and induction of pro-apoptotic factors [60]. Ivermectin, a member of the avermectin family, exhibits antitumor activity by causing oxidative stress and mitochondrial damage. In GB models, it has been observed to increase the apoptotic phenomena through a caspase-dependent mechanism and the inhibition of angiogenesis [61].

As previously reported, multiple mitochondrial functions and, in particular, oxidative phosphorylation, represent the main sources of energy for the development and proliferation of brain gliomas. A recent experimental study has evaluated the pharmacological activity of GA mitochondrial matrix inhibitor (gamitrib) [62]. This drug inhibits heat shock protein 90 (HSP90) which plays a key role in regulating the metabolic switch between oxidative phosphorylation and aerobic glycolysis [63]. The research carried out, using multiple glioma cell lines, has demonstrated the inhibition of mitochondrial energy production processes [62]. In a recent, interesting, experimental study, the efficacy of Gboxin, an inhibitor of oxidative phosphorylation obtained through the inhibition of the F0F1 ATPase complex V in mitochondria, was evaluated. The use of this compound is, however, limited by its instability, its short half-life, and its inability to cross the BBB. The authors, therefore, created a cancer cell–mitochondria hybrid membrane camouflaged ROS-responsive nanoparticle loaded with Gboxin (HM-NPs@G) to allow the release of the drug only in GB mitochondria [64]. The release of Gboxin, in tumor cells, interrupts the activity of ATP synthase at the mitochondrial inner membrane, causing the dysregulation of electron transport mechanisms, energy production mechanisms, and, finally, mitochondria-mediated apoptosis [65]. The results obtained highlighted a greater accumulation of the pharmacological compound in the tumor tissue and the inhibition of proliferation in U87MG and human-derived GB stem cells [64].

The validity of plant-derived phytochemicals has been demonstrated in the treatment of brain gliomas. These compounds can act directly on the mitochondrial components or, indirectly, on metabolic alterations due to mitochondrial dysfunctions [66]. Curcumin is a natural extract obtained from the plant, turmeric. In glioma cell lines, an increase in mitochondrial-mediated apoptosis has been observed through the release of cytochrome C and the increase in caspases 8, 9, and 3 [67]. Wang et al. structured a pharmacological compound consisting of berberine (extract of Chinese herbs) and glucose-responsive nanoparticles. The drug, tested on glioma cell lines, triggered the disappearance of the mitochondrial crista, the increase in ROS production, the increase in the production of the proteins (Bax, Bcl-2) involved in apoptosis processes, and, finally, cell cycle arrest [68]. In a recent experimental study, the authors demonstrated the greater efficacy of curcumin and berberine used synergistically. In glioma cell lines, the inhibition of the PI3/Akt/mTOR pathway, cellular DNA damage, increased ROS generation, alterations of and consequent reduction in the mitochondria within cells, and increased apoptosis were observed [69]. Aloe is a natural extract of *A. Vera*. Through experimental research, on glioma cell lines, the efficacy of aloe in inducing mitochondrial apoptosis by altering the membrane potential of mitochondria has been demonstrated [70]. Furthermore, it can upregulate the genes involved in mitochondrial apoptotic mechanisms and mitochondrial dynamics [71]. Quercetin is a flavonoid present in fruits and vegetables. Through studies on glioma cell lines, the induction of apoptotic mechanisms has been observed by stimulating the activity of caspases 9 and 3 [72]. Zhu et al., recently, evaluated the action of bufotalin as a potential treatment for GB. The researchers demonstrated, on tumor cell lines, the efficacy of bufotalin to inhibit AKT phosphorylation causing an accumulation of ROS inside the neoplastic cell and consequent alteration of mitochondrial functionality [73].

**Table 3 medicina-60-01888-t003:** Schematic representation of pharmacological compounds and the related potential mitochondrial targets and pathways.

Pharmacological Compounds	Mitochondrial Target	Pathway	Reference
Atovaquone	Inhibits Complex III	Oxidative phosphorylation	[57]
LON	Alters ATP production	Oxidative phosphorylation	[58]
Menadione/ascorbate	Mitochondrial superoxide system	Regulation of the redox state of the cell	[59]
YM155	Inhibits BCL-xL	Apoptosis and regulation of the cell cycle	[60]
Ivermectin	↑ Caspases protein production	Apoptosis and regulation of the cell cycle	[61]
Gamitrib	Inhibits HSP 90	Oxidative phosphorylation	[62,63]
Gboxin	Inhibits FOF1 ATPase	Oxidative phosphorylation	[64,65]
Curcumin	↑ Cyt C, Caspase 8, 9, 3	Apoptosis and regulation of the cell cycle	[67]
Berberine	↓ Mitochondrial crista, ↑ ROS, ↑ Bax, Bcl-2	Oxidative phosphorylation, regulation of the redox state of the cell, apoptosis, and regulation of the cell cycle	[68,69]
Aloe	Instability of the mitochondrial membrane potential	Oxidative phosphorylation, regulation of the redox state of the cell	[70]
Quercetin	↑ Caspase 9, 3	Apoptosis and regulation of the cell cycle	[72]
Bufotalin	Inhibits AKT	Apoptosis and regulation of the cell cycle	[73]

TMZ, the chemotherapeutic agent most used in the treatment of GB, must be correlated with high levels of ROS to exert its function. The ROS levels are dependent on the mitochondrial pool of glial tumor cells present in the glycolytic-dependent metabolic pathways, showing a phenotype resistant to apoptosis while O_2_-dependent metabolic pathways are sensitive to oxidative stress. GB is a mass of tumor cells that presents intracellular heterogeneity associated with a condition of mitochondrial heteroplasmy. Therefore, in this condition, cells reside intrinsically that down-regulate the action of the respiratory chain complexes in favor of the glycolytic pathway, favoring the survival of that pool of cells resistant to antitumor drugs, such as TMZ, responsible for chemo-resistance [74,75]. The effectiveness of antitumor therapy also depends on the tumor microenvironment that plays a major role in the growth and spread of tumor cells. Specifically, the study by Spees et al. [76] has documented, as happens among bacteria, the phenomenon of conjunction, that human mesenchymal stem cells can transfer functioning mitochondria to cells with non-functioning mitochondria, unable to ensure aerobic respiration, through a structure called tunneling nanotubes (TNTs). This demonstrates how, in unfavorable conditions, tumor cells, through their intrinsic plasticity, can respond to such circumstances not only by modifying their energy metabolism but also, and above all, their genetic makeup, through the formation of intercellular bridges. It has been observed that intercellular exchange is not only “linear” but can occur through the production of exosomes. Specifically, this inter-signaling phenomenon occurs in GB tumor cells that produce, release, and transmit into the microenvironment exosomes containing oncogenic messages, such as the proteins involved in mitochondrial functions, determining the so-called oncogenic field effect [77]. Burger et al. formulated a compound named tyrphostin AG17 [NSC 242557, (3,5-di-tert-butyl-4-hydroxybenzylidene) malononitrile]. This compound had both a lipophilic and a cationic component. This peculiar characteristic allowed the drug to cross the mitochondrial membranes and reside in the anionic environment of the mitochondrion. Exposure of glioma cell lines to this drug resulted in an increase in the sensitivity to carmustine [78].

## 6. Discussion and Conclusions

To date, despite the technological innovations and continuous advances in the biomolecular field, there is no optimal treatment for malignant brain gliomas. The most commonly adopted treatment strategy is the surgical approach followed by radiotherapy and chemotherapy. Nevertheless, the prognosis of these patients remains negative, achieving only a prolongation of survival. Modern target therapies, although conceptually very valid, have not yet had truly satisfactory results. Neoplastic progression is regulated by numerous biomolecules and pathways. It is likely that inhibiting or stimulating a single target may not be sufficient to slow and/or inhibit tumor growth [13]. Recently, studies of nanotechnology applied to medicine have demonstrated their effectiveness also in the treatment strategies of malignant brain gliomas. Nanoparticles, thanks to their physical-chemical peculiarities (size, shape, electrical charge), can cross the BBB. Furthermore, if appropriately engineered, they can selectively reach the neoplastic cells, releasing the drug and reducing the toxic effects on healthy cells.

The process of gliomagenesis presents peculiarities that distinguish the development and progression of glial tumors. The ability to evade apoptosis, making neoplastic cells capable of unlimited replication, the possibility of generating energy sources, using pathways that allow the production of ATP in oxygen-poor environments, and the ability to modify the tumor microenvironment, making it suitable for tumor development, highlight how the mitochondrion has a primary function in carrying out these mechanisms. Therefore, in light of the multiple mitochondrial functionalities (production of ATP, evasion of apoptosis, retrograde signaling, and effector of vital cellular processes), it appears intriguing to want to consider this organelle as a potential target in new treatment strategies for gliomas. In this study, we reiterated the key role shown by mitochondria in the processes of proliferation and progression of malignant brain gliomas. It seems, by now, established that mitochondria, during the processes of carcinogenesis and gliomagenesis, undergo a sort of reprogramming. This key mechanism would favor the neoplastic transformation of cells allowing them to proliferate in the tumor microenvironment. The exact process is still not fully understood, which greatly limits the search for new pharmacological compounds. However, other obstacles limit the effectiveness of these new strategies. The presence of the mitochondrial membrane, cellular heterogeneity, drug resistance, and the tumor microenvironment are tough obstacles to overcome, limiting the effectiveness of such treatments [68]. Furthermore, the results of these new therapeutic strategies have been obtained from experiments on glioma cell lines or tumor models, and clinical application is still lacking. In light of these observations, further research is needed to better understand the mechanisms that regulate the processes involving mitochondria in gliomagenesis and to structure new, more, specific drugs capable of inhibiting and/or limiting mitochondrial activity.

We have reported in the text, the potential effectiveness of natural plant extracts. These compounds show, in experimental studies, a real effectiveness in limiting mitochondrial activity. Their validity is, however, limited by the poor solubility and the difficulty in crossing the BBB. In this case, the nanoparticles, used as carriers, cross the BBB, reaching the tumor cells. Furthermore, the nanoparticles, if appropriately engineered, bind selectively to the glial cells, releasing the drug capable of altering and/or inhibiting mitochondrial functions [79].

New and more specific treatment strategies are needed. In an interesting study, the authors hypothesize that oxaloacetate can inhibit human lactate dehydrogenase A in glial cells and reverse the Warburg effect [80]. This would result in the restoration of correct mitochondrial activity. Furthermore, it has also been highlighted that the association between oxaloacetate and TMZ would lead to a prolongation of survival [81]. The synergistic use of drugs capable of inhibiting mitochondrial activity and chemotherapeutics or other pharmacological compounds capable of interacting with key targets of gliomagenesis could represent a valid alternative to the known strategies.

The research will have to further investigate all the mechanisms still not clarified to provide a broader scenario of targets and, consequently, more specific and effective drugs.

## Figures and Tables

**Figure 1 medicina-60-01888-f001:**
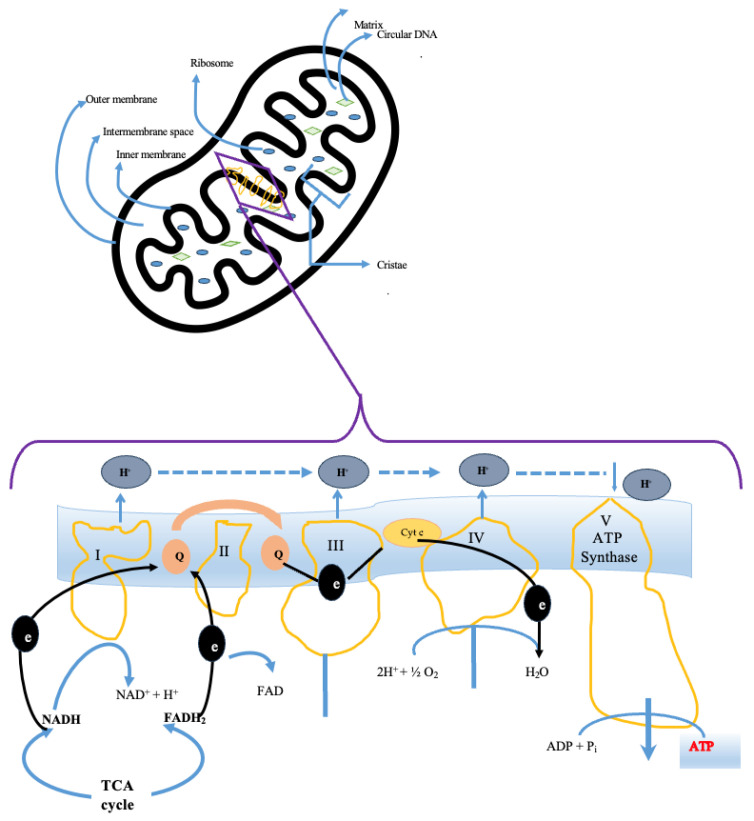
Schematic representation of mitochondrial structures and molecular pathways involved in ATP production.

**Table 1 medicina-60-01888-t001:** Mitochondrial complexes in oxidative phosphorylation.

Complex I NADH Dehydrogenase	Complex II Succinate Dehydrogenase	Complex III Cytochrome c Oxidoreductase	Complex IV Cytochrome c Oxidase	Complex V ATP Synthase
mtDNA 7	mtDNA 0	mtDNA 1	mtDNA 3	mtDNA 2
ntDNA 35	ntDNA 4	ntDNA 10	ntDNA 10	ntDNA 14

**Table 2 medicina-60-01888-t002:** Schematic representation of the phenomenon of heteroplasmy and of mitochondrial plasticity in tumor-initiating cells (TICs).

TIC and mitochondrial plasticity
Regression of mitochondrial density, distribution, and ultrastructural aspects
Mitochondrial clearance
Reduction in subunit expression complex I and IV
Increase in subunit expression complex II, III, V
Epigenetic modifications
Remodeling metabolic pathways
Asymmetrical segregation of mitochondria (heteroplasmy condition)
Cataplerosis

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
