# Peer review of "Mitochondrial Dysfunction: Effects and Therapeutic Implications in Cerebral Gliomas"

_medicina, 2024, doi:10.3390/medicina60111888_

Round 1
Reviewer 1 Report
Comments and Suggestions for Authors
Overall:
This manuscript reviews the literature on organizing mitochondria as a potential therapeutic implication in cerebral gliomas and may become a diagnostic direction. Although this is an exciting and timely topic for review, the manuscript lacks integration and may be disjointed, possibly due to input from multiple authors. In just one sentence, many vital studies are mentioned broadly and would benefit significantly from adding details (such as whether glioma drugs are currently designed to target mitochondria). Adding details and reorganizing comments (consolidating redundant topics) will also help guide readers. Additional questions are listed below.
Major review:
1. The author should add a paragraph explaining why he considers the target mitochondria of glioma necessary.
2. The passages on mitochondrial activity, ATP Production, and apoptosis can be condensed to increase the difficulties encountered in Therapeutic Strategies.
Minor review
1. In Table 1, reorganized Table 1 is too simple; please add references.
2. Add the latest glioma and glioma drugs for target mitochondria detail in the new table.
3. To suggest a summary figure to point to the potential therapeutic pathway in mitochondria
4. The figure 1 could be removed.
Author Response
We thank the Reviewer for your suggestions. We have carefully read the reviewer’s comments and have revised our manuscript accordingly.
- This manuscript reviews the literature on organizing mitochondria as a potential therapeutic implication in cerebral gliomas and may become a diagnostic direction. Although this is an exciting and timely topic for review, the manuscript lacks integration and may be disjointed, possibly due to input from multiple authors. In just one sentence, many vital studies are mentioned broadly and would benefit significantly from adding details (such as whether glioma drugs are currently designed to target mitochondria).
- Following the reviewer's suggestion, we have reread the text. We have reduced some sections and subsections (Mitochondrial Activity, ATP Production, Apoptosis) and, at the same time, we have inserted new concepts.
- The author should add a paragraph explaining why he considers the target mitochondria of glioma necessary.
- Following the reviewer's observation, we have inserted in the discussion section the following concepts in the revised text: “The process of gliomagenesis presents peculiarities that distinguish the development and progression of glial tumors. The ability to evade apoptosis, making neoplastic cells capable of unlimited replication, the possibility of generating energy sources, using pathways that allow the production of ATP in oxygen-poor environments, the ability to modify the tumor microenvironment, making it suitable for tumor development, highlight how the mitochondrion has a primary function in carrying out these mechanisms. Therefore, considering the multiple mitochondrial functionalities (production of ATP, evasion of apoptosis, retrograde signaling, and effector of vital cellular processes) it appears intriguing to want to consider this organelle as a potential target in new treatment strategies for gliomas”.
- The passages on mitochondrial activity, ATP Production, and apoptosis can be condensed to increase the difficulties encountered in Therapeutic Strategies.
- Following the reviewer's suggestions, we have shortened the "Mitochondrial Activity" section and rewritten the "Therapeutic Strategies" section in the revised text:
Therapeutic Strategies
To date, the treatment of malignant gliomas represents a difficult challenge due to cellular and molecular heterogeneity. In the management of these patients, several important factors must be considered, such as histotype of the lesion, location and volume, extent of surgical resection and residual volume, Karnofsky Performance Score, and positivity of specific markers [9]. Furthermore, the efficacy of current chemotherapeutic treatments is limited by the presence of the blood-brain barrier (BBB) and, the presence of genetic mutations (IDH1, PTEN, and 1p19q codeletion), which can interfere with the response to these treatments. Gliomas, particularly high-grade gliomas, consist of multiple subtypes of tumor cells with diverse genetic profiles, making it difficult to target the tumor comprehensively with a single therapy [10]. Epigenetic modifications, enzymes, and noncoding RNAs are often cell-type specific and can aid in the identification of the cell of origin. Koso et al. evidenced that the cell of origin in some GB is an astroglial-like cell and that the originating mutations can occur in neural stem cells [11]. The immunosuppressive tumor microenvironment of gliomas, characterized by the presence of regulatory T cells, tu-mor-associated macrophages, and microglia, reduces the efficacy of immunotherapy and complicates treatment efforts [12].
The most commonly adopted treatment of gliomas is multimodal, including surgical resection, radiotherapy, and chemotherapy. Improvements in surgical technique, including intra-operative mapping of eloquent areas and the use of fluorescent dyes that are helpful in the detection of tumor borders, may only offer some benefits in prolonging survival. Surgical treatment is invasive but represents the first choice for gliomas due to difficulties arising in early-stage detection. However, radical resection is not always achievable, both due to the extensive infiltration of the tumor and as an attempt to preserve the functional areas. Furthermore, after surgical treatment, neoplastic cells tend to recur immediately, and in 95% of cases, the recurrence begins from the areas adjacent to the margins of the surgical resection [13-14]. Furthermore, when the tumor is in deep structures (diencephalon, midbrain, corpus callosum) it is preferable to perform a biopsy to obtain a histological diagnosis of the tumor. Radiation therapy and chemotherapy are non-invasive options often used as adjuvant therapy but may also be effective for curing early-stage tumors. Radiotherapy is burdened by important side effects such as post-radiation leukoencephalopathy, nerve damage, hair loss, vomiting, infertility, and skin rash [15]. The BBB remains a significant obstacle for many therapies, including chemotherapeutics and targeted therapies, as it restricts the penetration of large molecules into the brain [13, 15]. BBB is a physical barrier that protects the brain from the passage of drugs, neurotoxins, and invading organisms and regulates the passage of nutrients between the systemic circulation and the brain. Most of the chemotherapeutic drugs used have reduced solubility, are combined with toxic solvents, and, showing a reduced specificity for the targets, can also cause damage to healthy cells.
Gliomagenesis is a multifactorial process involving a large number of molecules. ECM proteins, proteases, cell adhesion molecules, and their related signaling pathways show a vital role in glioma proliferation [16]. The blockage of activation of oncogenic pathways, either at the ligand receptor interaction level or by inhibiting downstream signal transduction pathways, could slow down tumor progression. Approximately 50% of GBs show amplification or mutation in the epidermal growth factor receptor (EGFR), making this a promising target for therapies such as tyrosine kinase inhibitors (TKIs) [17]. Some gliomas, especially pediatric gliomas, may harbor proto-oncogene B-Raf (BRAF) mutations. Inhibitors targeting the BRAF V600E mutation are being explored [18]. Phosphoinositide 3-kinase (PI3K)/protein kinase B (PKB or AKT)/mammalian target of rapamycin (mTOR) pathway is often dysregulated in gliomas, leading to unregulated cell growth and survival. Inhibitors targeting this pathway, such as (mTOR) inhibitors, are under investigation [19]. The complexity and crosstalk between signal transduction pathways limit the potential efficacy of targeting a single receptor or molecule. Since single-agent therapies have shown no significant benefit, it is critical to begin designing rational combinations. There is a need for combinatorial approaches or novel delivery mechanisms to overcome resistance and enhance the effectiveness of these therapies.
Immunotherapy is becoming a prominent focus in glioma research, particularly for high-grade gliomas. Oncolytic virotherapy using engineered viruses, chimeric T cell receptors (CAR T)-cell therapy targeting glioma-specific antigens like epidermal growth factor receptor (EGFRvIII), and bispecific antibodies represent innovative immunotherapeutic strategies. Despite some promising early-phase results, these approaches still require further investigation to improve efficacy [20]. Each of these approaches represents efforts toward more personalized and effective glioma treatment strategies. Techniques like suicide gene therapy (which delivers genes to tumor cells that trigger their death) and tumor suppressor gene therapy are being explored to provide more targeted treatment. Even their clinical efficacy is still under evaluation [21]. The advancement of gene-editing tools like clustered regularly interspaced short palindromic repeats/protein 9 (CRISPR/Cas 9) [22] offers new possibilities for correcting genetic mutations associated with glioma growth. Epigenetic therapies, such as drugs targeting DNA methylation or histone modifications, are also being investigated to modulate gene expression in glioma cells. However, these strategies are still in early research phases [23]. Numerous studies have also highlighted the pivotal role of mitochondria‑related genes (MRGs) in the initiation and progression of GB. However, the specific contributions of MRG coding proteins to GB pathology remain incompletely elucidated. The identification of prognostic MRGs in GB holds promise for the development of personalized targeted therapies and the enhancement of patient prognosis.
Vaccine therapies for gliomas aim to stimulate the immune system to recognize and attack tumor cells. Vaccines based on specific glioma-associated antigens, like the EG-FRvIII mutation, are being tested in clinical trials. Although early results are promising, there are a lot of trial still ongoing to prove if there is a significant improvement in overall survival [24].
An emerging approach in glioma treatment is theranostics, which combines diagnostic and therapeutic capabilities into a single platform. For instance, nanoparticles can be engineered to deliver therapeutic agents while simultaneously allowing for imaging of tumor progression or treatment response. This personalized medicine approach holds the potential to improve glioma management [25].
Minor review
- In Table 1, reorganized Table 1 is too simple; please add references.
- In agreement with the reviewer, we have removed Table 1 from the revised text.
- Add the latest glioma and glioma drugs for target mitochondria detail in the new table.
- We have therefore inserted a new table (table 3), in which we report the pharmacological compounds, the related potential mitochondrial targets and pathways, and the specific references.
|
Pharmacological Compounds |
Mitochondrial Target |
Pathway |
Reference |
|
Atovaquone |
Inhibits Complex III |
Oxidative phosphorylation |
46 |
|
LON |
Alters ATP production |
Oxidative phosphorylation |
47 |
|
Menadione/ascorbate |
Mitochondrial superoxide system |
Regulation of the redox state of the cell |
48 |
|
Ivermectin |
Caspases protein production |
Apoptosis and regulation of the cell cycle |
49 |
|
YM155 |
Inhibits BCL-xL |
Apoptosis and regulation of the cell cycle |
50 |
|
Gamitrib |
Inhibits HSP 90 |
Oxidative phosphorylation |
51, 52 |
|
Gboxin |
Inhibits FOF1 ATPase |
Oxidative phosphorylation |
53 |
|
Curcumin |
Cyt C, Caspase 8,9,3 |
Apoptosis and regulation of the cell cycle |
56 |
|
Berberine |
¯ Mitochondrial crista, ROS, Bax, Bcl-2 |
Oxidative phosphorylation, regulation of the redox state of the cell, apoptosis, and regulation of the cell cycle |
57 |
|
Aloe |
Instability of mitochondrial membrane potential |
Oxidative phosphorylation, regulation of the redox state of the cell |
59 |
|
Quercetin |
Caspase 9,3 |
Apoptosis and regulation of the cell cycle |
61 |
|
Bufotalin |
Inhibits AKT |
Apoptosis and regulation of the cell cycle |
62 |
Table 3: Schematic representation of pharmacological compounds and the related potential mitochondrial targets and pathways.
- To suggest a summary figure to point to the potential therapeutic pathway in mitochondria
- We have therefore inserted a new table (table 2), in which we report the pharmacological compounds, the related potential mitochondrial targets and pathways, and the specific references.
Figure 1 could be removed.
- In agreement with the reviewer, we have removed Figure 1 in the revised text.

Reviewer 2 Report
Comments and Suggestions for Authors
The title of the manuscript and the abstract is very interesting and well phrased, however the contents need substantial improvements. It would be easier to comment if the authors had put line numbers. In the sentence "Mitochondria perform various functions such as cellular bio-energetic production, regulation of apoptosis and retrograde signaling" please discuss a bit more about retrograde signaling. The sentence "Gliomas conventional treatment consists of surgical resection, followed by, radiation therapy combined to chemotherapy with TMZ and then TMZ alone." appears in multiple places in the manuscript and seems redundant information, please correct. In the first paragraph of page 4, please cite the references used and include a good number of references as this section of the manuscript related to the title of the review directly. In the legend of Figure1 the spelling of mitochondrion is wrong, please correct. In this context, I don't see a reason for including this basic image of mitochondria at all, rather include a concise figure for the molecular pathways involved in ATP production and their connectivity with each other. A concise figure/table containing information on heteroplasmy and TIC should be included in the revised version of the manuscript. One last suggestion that would really help in improving the quality of this article is a figure clearly illustrating the role of various MRG's in mitochondrial Complexes (with special emphasis on complex 1 and bioenergetic levels) and their dysfunctions due to mutation in the TICs and other glial cells in brains of patients with gliomagenesis.
Author Response
We thank the Reviewer for your suggestions. We have carefully read the reviewer’s comments and have revised our manuscript accordingly.
- In the sentence "Mitochondria perform various functions such as cellular bio-energetic production, regulation of apoptosis and retrograde signaling" please discuss a bit more about retrograde signaling.
- Following the reviewer's suggestion, we have removed the sentence from the "Introduction" section. We have therefore inserted the following information in the revised text, in the "Mitochondrial Dynamics and Retrograde Signaling" subsection: “Mitochondrial retrograde signaling is a peculiar flow of data between the mitochondrion and the nucleus that occurs even in conditions of altered mitochondrial functionality. Generally, the loss of membrane potential, due to alterations of the respiratory chain components or to mutations and/or alterations of mtDNA, triggers mitochondrial retrograde signaling. This mechanism should allow the restoration of correct mitochondrial functionality [41]”.
- The sentence "Gliomas conventional treatment consists of surgical resection, followed by, radiation therapy combined to chemotherapy with TMZ and then TMZ alone." appears in multiple places in the manuscript and seems redundant information, please correct.
- Following the reviewer's observation, we have removed redundant information in the revised text.
- In the first paragraph of page 4, please cite the references used and include a good number of references as this section of the manuscript related to the title of the review directly.
- Following the reviewer's suggestion, we have added further references to the revised text.
- In the legend of Figure 1 the spelling of mitochondrion is wrong, please correct. In this context, I don't see a reason for including this basic image of mitochondria at all, rather include a concise figure for the molecular pathways involved in ATP production and their connectivity with each other.
- In agreement with the reviewer, we have removed Figure 1 from the revised text. We have consequently added a new Figure 1 to the revised text.
Figure 1: Schematic representation of mitochondrial structures and molecular pathways involved in ATP production
- A concise figure/table containing information on heteroplasmy and TIC should be included in the revised version of the manuscript.
- In the revised text we have added a new table (Table 2). In this table, we have reported additional data regarding the phenomenon of heteroplasmy and mitochondrial plasticity in tumor-initiating cells (TICs).
|
TIC and mitochondrial plasticity |
|
Regression of mitochondrial density, distribution, and ultrastructural aspects |
|
Mitochondrial clearance |
|
Reduction of subunit expression complex I and IV |
|
Increase of subunit expression complex II, III, V |
|
Epigenetic modifications |
|
Remodeling metabolic pathways |
|
Asymmetrical segregation of mitochondria (heteroplasmy condition) |
|
Cataplerosis |
Table 2: Schematic representation of the phenomenon of heteroplasmy and of mitochondrial plasticity in tumor-initiating cells (TICs).
- One last suggestion that would really help in improving the quality of this article is a figure clearly illustrating the role of various MRG's in mitochondrial Complexes (with special emphasis on complex 1 and bioenergetic levels) and their dysfunctions due to mutation in the TICs and other glial cells in brains of patients with gliomagenesis.
- Following the reviewer's suggestion, we have added a new figure (Figure 1) and a new table (Table 1) to the revised text. In Figure 1, we report the schematic representation of mitochondrial structures and molecular pathways involved in ATP production. In Table 1, we report the mitochondrial complexes that play a key role in oxidative phosphorylation. We also report the number of mitochondrial and nuclear DNAs involved in each complex.
|
Complex I NADH dehydrogenase |
Complex II Succinate dehydrogenase |
Complex III Cytochrome c oxidoreductase |
Complex IV Cytochrome c oxidase |
Complex V ATP Synthase |
|
mtDNA 7 |
mtDNA 0 |
mtDNA 1 |
mtDNA 3 |
mtDNA 2 |
|
ntDNA 35 |
ntDNA 4 |
ntDNA 10 |
ntDNA 10 |
ntDNA 14 |
Table 1: Mitochondrial complexes in oxidative phosphorylation.

Round 2
Reviewer 1 Report
Comments and Suggestions for Authors
The authors have addressed all questions in this manuscript. I suggest the authors have adequately addressed my concerns in the first review. Therefore, this manuscript could be accepted.
Comments on the Quality of English LanguageThe English of the article has improved, and it is easy to read.
Reviewer 2 Report
Comments and Suggestions for Authors
The revision has been done satisfactorily and the manuscript can be accepted in present form.